# Scalable and Improved Algorithms for Individually Fair Clustering

**MohammadHossein Bateni**
Google Research
bateni@google.com

**Vincent Cohen-Addad**
Google Research
cohenaddad@google.com

**Alessandro Epasto**
Google Research
aepasto@google.com

**Silvio Lattanzi**
Google Research
silviol@google.com

## Abstract

We present scalable and improved algorithms for the individually fair $(p, k)$-clustering problem introduced by Jung et al. [2020] and Mahabadi and Vakilian [2020]. Given $n$ points $P$ in a metric space, let $\delta(x)$ for $x \in P$ be the radius of the smallest ball around $x$ containing at least $n/k$ points. A clustering is then called *individually fair* if it has centers within distance $\delta(x)$ of $x$ for each $x \in P$. In this work, we present two main contributions. We first present local-search algorithms improving prior work along cost and maximum fairness violation. Then we design a fast local-search algorithm that runs in $\tilde{O}(nk^2)$ time and obtains a bicriteria $(O(1), 6)$ approximation. Finally we show empirically that not only is our algorithm much faster than prior work, but it also produces lower-cost solutions.

## 1 Introduction

The $(p, k)$-clustering problems (with $k$-means, $k$-median and $k$-center as special cases) are widely used in many unsupervised machine-learning tasks for exploratory data analysis, representative selection, data summarization, outlier detection, social-network community detection and signal processing, e.g., Lloyd [1982], MacQueen [1967], Chawla and Gionis [2013], Kleindessner et al. [2019], Zhang et al. [2007], Bóta et al. [2015]. With such ubiquity of applications, it is fundamental to design fair algorithms for such problems. In this paper we focus on the notion of individually fair clustering Jung et al. [2020], which *combines* the $\ell_p$ cost objective with a $k$-center-based concept of fairness: *A minimum level of treatment should be guaranteed for every data point*. To better understand this formulation consider the case in which centers were chosen randomly. In this case any subset of $n/k$ points would expect to include one center. So each point desires to be assigned to a center among its $n/k$ closest points. This notion can be captured by considering a different radius $\delta(x)$ for each $x$ in the dataset and by adding the constraint that there should be a center within distance $\delta(x)$ for each $x$. Satisfying such constraints amounts to (a special case of) the priority $k$-center problem Plesník [1987], Alamdari and Shmoys [2017].

Shortly after Jung et al. [2020] proposes this problem and presents a a 2-approximation for it, Mahabadi and Vakilian [2020] generalizes it to an optimization setting where an $\ell_p$ norm cost function (such as $k$-means or $k$-median) is optimized within the space of individually fair solutions. In fact, they devise a local-search algorithm with bicriteria $(84, 7)$ approximation (for $p = 1$, that is $k$-median).

In the past year, several attempts have been made to improve these results, theoretically and practically. Chakrabarty and Negahbani [2021] uses LP rounding to improve the guarantee to $(2^{p+2}, 8)$, i.e.,

2022 Trustworthy and Socially Responsible Machine Learning (TSRML 2022) co-located with NeurIPS 2022.

cost guarantee of 16 for $k$-means and 8 for $k$-median. In simultaneous work, Humayun et al. [2021] presents an SDP-based algorithm (without performance or runtime analysis), and Vakilian and Yalçiner [2022] presents LP-based bicriteria guarantees $(16^p + \epsilon, 3)$ for any $p$ and $(7.081 + \epsilon, 3)$ for $p = 1$.[1]

Three of the above—with the exception of Vakilian and Yalçiner [2022]—present experimental studies corroborating the usefulness of the new algorithms. However, a major limitation of these algorithms is their running times, having an exponent of at least 4 for the number of points $n$, making them impractical for real-world datasets of interest. Therefore, as we explain in our empirical studies, prior experiments ran only on small datasets of size at most 1000.

**Our contributions.** We have two main contributions. First we present a local-search algorithm for the problem, complementing the state-of-the-art guarantees of prior published work. As in previous work, we focus on $p = 1$ ($k$-median) for simplicity: Setting $\gamma = 6$ gives a bicriteria guarantee of $(3, 7)$, improving along both objectives the $(84, 7)$ guarantee from Mahabadi and Vakilian [2020], and the $(8, 8)$ guarantee from Chakrabarty and Negahbani [2021]. As such, our algorithm is not comparable to the $(7.081, 3)$ guarantee of Vakilian and Yalçiner [2022] or the $(8, 4)$ guarantee of Alamdari and Shmoys [2017] (for the uniform radius special case), since our algorithm achieves a better bound for the cost but a worse bound for the fairness constraint.

**Theorem 1.1.** *Let $\gamma > 4$ and $\varepsilon > 0$. Assuming the problem is feasible (i.e., there exists an individually fair solution), there is a polynomial-time algorithm for individually fair $k$-median with bicriteria guarantee $(\alpha_\gamma, \gamma + 1)$, where $\alpha_\gamma = 3 + O(\varepsilon)$ for $\gamma \geq 6$ and $\alpha_\gamma = \frac{2 + \frac{4}{\gamma - 2}}{2 - \frac{4}{\gamma - 2}} + O(\varepsilon)$ for $6 > \gamma > 4$.*

As mentioned, the previous algorithms (and our new result above) have poor runtime guarantees, making them impractical for real-world datasets of interest. In fact, prior experiments focused on samples of 1000 points only. To address this shortcoming, we design a fast local-search algorithm using ideas from the algorithm above as well as from the fast $k$-means algorithm of Lattanzi and Sohler [2019]. We focus on $k$-means, which is more commonly used in practice.

**Theorem 1.2.** *There is an $\tilde{O}(nk^2)$-time algorithm for individually fair $k$-means with a 6-approximation for radii and an $O(1)$-approximation on costs.*

This is the algorithm we implement for our empirical study. It uses local search with swaps of size one, and incorporates ideas similar to Lattanzi and Sohler [2019] to find good swaps quickly. Whereas our algorithm takes less than 60 minutes to process half a million points, prior methods hit the one-hour mark[2] with 4000–15000 points; see Figure 1a. Despite the worse approximation guarantees, we observe in Section 5 that this algorithm outperforms prior algorithms on cost and fairness objectives.

Note that our algorithms, and some of the prior work Mahabadi and Vakilian [2020], Chakrabarty and Negahbani [2021], Vakilian and Yalçiner [2022], work for any vector $\delta$ of radius bounds. The individually fair setting is only one application.

**Additional Related work.** There is an extensive literature on group fairness, where the goal is to curb under- and over-representation in certain slices of the data (say, sensitive groups based on gender or age group) Chierichetti et al. [2017], Rösner and Schmidt [2018], Bera et al. [2019], Ahmadian et al. [2019, 2020b,a]. Another line of work concerns two generalizations of $k$-clustering problems to $\ell_p$ norm and ordered median objectives Byrka et al. [2018], Chakrabarty and Swamy [2018, 2019], Kalhan et al. [2019]: Create a (non-increasingly) sorted vector out of the distances of points to their closest centers, and aim to minimize either the $\ell_p$ norm of this vector or the inner product of the vector with some given weight vector $w$. Note that $p = 1, 2, \log n$ yields $k$-median, $k$-means and $k$-center through the first generalization, whereas $w = (1, 1, \ldots, 1)$ and $w = (1, 0, \ldots, 0)$ yield $k$-median and $k$-center objectives through the latter. Chlamtác et al. [2022] combines the two generalizations into the notion of $(p, q)$-Fair Clustering problem, which is also a generalization of Socially Fair $k$-Median and $k$-Means Ghadiri et al. [2021], Abbasi et al. [2021], Makarychev and Vakilian [2021]. Some of the above results are also motivated from the standpoint of solution *robustness*—the main motivation

---

[1]We note that the original version of their manuscript (simultaneous to our work) presented a $(8 + \epsilon, 3)$ guarantee, but they updated their results with a better guarantee only in March this year.

[2]Notice that the LP- or SDP-based algorithms require $\Omega(n^2)$ space, so it is unclear whether those algorithms can scale to larger datasets even with more runtime allowance, as they run out of memory around the same data size.

stated in Humayun et al. [2021]. The widely popular $k$-means clustering implicitly assumes certain uniform Gaussian distributions for the data Raykov et al. [2016], and is known to be sensitive to sampling biases and outliers Wang et al. [2020]. Beyond enforcing individual fairness or cluster-level consistency constraints (the focus of this work), researchers have tackled the above problem from various angles such as resorting to kernel methods Dhillon et al. [2004], adding regularization terms Georgogiannis [2016], and using trimming functions Georgogiannis [2016], Deshpande et al. [2020], Dorabiala et al. [2021].

**Organization.** We start with some preliminaries in Section 2. Then in Section 3 we present our local-search algorithm with improved cost and fairness guarantees. Section 4 outlines the faster local-search algorithm, and proves bounds on its performance. Finally we present results of our empirical study in Section 5.

## 2 Preliminaries

Let $(X, \text{dist})$ be a metric space, where $X$ is a set of points and dist a distance function between the points in $X$. We define the distance between a point $p$ and a set of points $C$ as $\text{dist}(p, C) = \min_{c \in C} \text{dist}(p, c)$; if the set $C$ is empty we define the distance to be $\infty$. Let $\Delta = \max_{p,q} \text{dist}(p, q) / \min_{p \neq q} \text{dist}(p, q)$ denote the *aspect ratio* of the instance.

**Problem definition.** Given a metric space $(X, \text{dist})$, the input to our problem is a tuple $(A, C, k, \delta)$, where $A \subseteq X$ is a set of points of the metric space called *clients*, $C \subseteq X$ is a set of points of the metric space called *candidate centers*, $k$ is a positive integer and $\delta : A \mapsto \mathbb{R}_+$ is the desired serving cost or *radius* of points. The goal is to output a set $S \subseteq C$ that minimizes $\sum_{a \in A} \text{dist}(a, S)$ under the constraints that $|S| \leq k$ and $\forall a \in A : \text{dist}(a, S) \leq \delta(a)$.

The element of a solution $S \subseteq C$ are called *centers* or *facilities*. Given a set $S$ of $k$ centers, let $\text{cost}(S)_1$ denote the $k$-median cost of the client set $C$ for the centers $S$, i.e., $\text{cost}(S)_1 = \sum_{c \in C} \text{dist}(c, S)$. Similarly, we define the cost of the client set $C$ for the centers $S$ for the $k$-means problem as $\text{cost}(S)_2 = \sum_{c \in C} \text{dist}(c, S)^2$. In both setting we denote with $\text{Opt}_k$ the cost of an optimum solution $S^*$. When it is clear from the context we will drop the index 1 or 2 from the notation for the cost.

A solution is an $(\alpha, \beta)$ bicriteria approximation if the $k$-median (or $k$-means) cost of the solution is at most $\alpha$ times that of the optimum, while the $k$-center (or fairness) constraint is violated by at most $\beta$.

---

**Algorithm 1:** SEEDING

---

**Require:** $A, \delta(\cdot), \gamma$
1:  $S = \emptyset$
2:  **while** $\exists p \in A : \text{dist}(p, S) > \gamma \delta(p)$ **do**
3:      $p^* \leftarrow \arg \min_{p' \in \{p \in A | \text{dist}(p,S) > \gamma \delta(p)\}} \delta(p')$
4:      $S \leftarrow S \cup \{p^*\}$
5:  **Output** $S$

---

We describe the seeding procedure outlined in Algorithm 1 to initialize our local-search approach.

**Lemma 2.1.** *If the problem is feasible, Algorithm 1 with parameter $\gamma > 2$ returns a set of points $S$ of size at most $k$ such that each point $p$ is at distance at most $\gamma \delta(p)$ from the closest point in $S$, i.e., $\forall p \in A : dist(p, S) \leq \gamma \delta(p)$.*

*Proof.* The proof is similar to the proof of correctness of Gonzales' algorithm for $k$-center Gonzalez [1985] and of Hochbaum and Shmoys' algorithm Hochbaum and Shmoys [1985]. Observe first that by feasibility, there cannot be $k + 1$ points $p'_1, \ldots, p'_{k+1}$ such that the balls $\delta(p'_i)$ are all pairwise disjoint (since otherwise the optimum solution would need $k + 1$ centers to satisfy the $\delta(p')$ constraints).

Let $p_1, \ldots, p_{k^*}$ be the sequence of points picked by the algorithm. We have that $\delta(p_i) \leq \delta(p_j)$ for any $i \leq j$. Note that at the end of the algorithm, each point $p$ is at distance at most $\delta(p)$ from one of $p_1, \ldots, p_{k^*}$ so what remains to be shown is that $k^* \leq k$. We claim that the collection of balls centered at the $p_i$ and of radii $\delta(p_i)$ are all pairwise disjoint and so if the problem is feasible, the algorithm does not return more than $k$ points (i.e.: $k^* \leq k$). Consider a pair $i, j$ and without loss of generality $i > j$. We have that $p_i$ is at distance at least $\gamma \delta(p_i)$ from $p_j$ by the definition of the algorithm. Since

$\gamma > 2$ and $\delta(p_i) \geq \delta(p_j)$, we have that $\delta(p_i) + \delta(p_j) \leq 2\delta(p_i) < \gamma\delta(p_i) \leq \text{dist}(p_i, p_j)$ and so the ball of radius $\delta(p_j)$ around $p_j$ cannot intersect the ball of radius $\delta(p_i)$ around $p_i$.

$\square$

---

**Algorithm 2:** Anchored Local Search Algorithm

---

**Require:** $A$, $C$, $\delta(\cdot)$, $k$ and parameters $\gamma, \varepsilon$
 1: $A_0 \leftarrow \text{SEEDING}(A, \delta(\cdot), \gamma)$
 2: $S_0 \leftarrow \bigcup_{a \in A_0} \{\arg\min_{x \in C} \text{dist}(x, a)\}$
 3: **if** $|S_0| > k$ **or** $\exists a \in A_0 : \delta(a) < \text{dist}(a, S_0)$ **then**
 4:    **Output** `infeasible`.
 5: Define each $p \in A_0$ as an *anchor point*, the ball $B(p)$ centered at $p$ of radius $\delta(p)$ as its *anchor zone*.
 6: **while** $\exists S' : |S'| \leq k, |S' \setminus S_0| + |S_0 \setminus S'| \leq 12/\epsilon, S' \cap B(p) \neq \emptyset \forall p \in A_0$, and $\text{cost}(S') < (1 - \varepsilon/k)\text{cost}(S_0)$ **do**
 7:    $S_0 \leftarrow S'$
 8: **Output** $S_0$

---

## 3  Improved approximation guarantees

In this section we focus on the $k$-median problem and this section is dedicated to the proof of Theorem 1.1. We analyze the performances of Algorithm 2.

We let $\text{OPT}(c) = \text{dist}(c, \text{OPT})$ and $S_0(c) = \text{dist}(c, S_0)$ respectively denote the ($k$-median) cost contribution of client $c$ in solution OPT and in the solution $S_0$ output by the algorithm.

*Proof of Theorem 1.1.* Let OPT denote the optimum solution, let $S_0$ be the solution output by the above algorithm. We start by arguing that all clients are at distance at most $(\gamma + 1)\delta$ from a center of $S_0$. Observe that since we assume the problem is feasible and by Lemma 2.1, each client is at distance at most $\gamma\delta$ from an anchor point.

Since $S_0$ is constrained to contain a center in each anchor zone, an immediate application of the triangle inequality implies that each client $c$ is at distance at most $(\gamma + 1)\delta(c)$ from the output solution. We thus turn to the analysis of the ($k$-median) cost of the solution.

We follow the approach to analyse local search with a little twist to handle anchor points presented in Gupta and Tangwongsan [2008], Cohen-Addad and Schwiegelshohn [2017].

Consider an anchor point $p$. By definition of the problem and the fact that it is feasible, we have that there exists at least one center of OPT at distance less than $\delta(p)$ from $p$. Let $\text{OPT}_{B(p)}$ be such a center (chosen arbitrarily among the ones at distance at most $\delta(p)$ from $p$), and call it an *anchor* center. Similarly, let $S_{B(p)}$ denote a center of $S_0$ in $B(p)$ (there must be one by definition of the algorithm).

We now turn to the analysis. Consider the following bipartite graph $G$ with the two sets of vertices being the centers of OPT and $S_0$. Create an edge from each non-anchor center of OPT to the closest center in $S_0$. Vertices of $S_0$ with positive degree are called *leaders*. For each anchor center $\text{OPT}_{B(p)}$ of OPT, create an edge to the closest center in $S_0$ that is located in the ball centered at $p$ and of radius $\frac{\gamma}{2}\delta(p)$ (since $\gamma \geq 2$, this is always possible).

Now create groups $(X_i, Y_i)$, where $X_i \subseteq S_0$ and $Y_i \subseteq \text{OPT}$ as follows. For each center $\ell \in S_0$, create a group $(\{\ell\}, Y)$ where $Y$ is the set of centers of OPT adjacent to $\ell$. Note that each center of OPT belongs to exactly one group. Then, for each anchor center $\text{OPT}_{B(p)}$, consider the associated anchor center of $S_0$, namely $S_{B(p)}$. If $S_{B(p)}$ has degree 0 and the group $(X, Y)$ containing $\text{OPT}_{B(p)}$ has $|Y| > 1$, then add $S_{B(p)}$ to $X$.

Finally, add the centers of $S_0$ not in any groups yet arbitrarily to the groups such that each group $(X, Y)$ has $|X| = |Y|$. Note that this is always possible since both the optimum and the local solution contain $k$ centers.

We finally partition the groups into *subgroups* that are pairs of centers $(X, Y)$, $X \subseteq S_0$ and $Y \subseteq$ OPT. We define the *size* of a subgroup $(X, Y)$ as $|X| + |Y|$. Our partition of the groups into subgroups will satisfy the following two properties: (1) each subgroup is of size at most $2/\varepsilon$; and (2) for each group $(X, Y)$ of size larger than $2/\varepsilon$, the leader $\ell_g$ of $(X, Y)$ does not belong to any subgroup. Any group $(X, Y)$ such that $|X| \leq 1/\varepsilon$ becomes a subgroup. Then, for any other group $g = (X, Y)$, let $\ell_g$ be the leader. Partition $X$ and $Y$ into subsets $X_1, X_2, \ldots$ and $Y_1, Y_2, \ldots$ each of size $1/\varepsilon$ (except possibly for one subset of $X$ and one subset of $Y$); define subgroups $(X_i, Y_i)$ such that $|X_i| = |Y_i|$. Furthermore, consider the subgroup $(X_j, Y_j)$ containing $\ell_g$ and pick uniformly at random a center $\ell_r$ in $X - X_j$ and replace $\ell_g$ by $\ell_r$ in $X_j$, namely the subgroup becomes $(X_j - \{\ell_g\} \cup \{\ell_r\}, Y_j)$. Note that this random process is only introduced for the analysis – not in the algorithm itself.

We now define the swaps by merging some subgroups. We define pairs $(X, Y)$ satisfying both (1) $|X| + |Y| \leq 12/\varepsilon$ and (2) the solution $S_0 - X \cup Y$ has one center in each anchor zone. By local optimality, we will thus have that the cost of $S_0 - X \cup Y$ is at least as large as $(1 - \varepsilon/k)$ times the cost of $S_0$.

Consider an anchor zone $B$. We want to make sure for any swap pair $(X, Y)$ if $S_{B(p)}$ is in $X$, then $\text{OPT}_{B(p)} \in Y$, ensuring feasibility. To do so, consider the subgroup containing $\text{OPT}_{B(p)}$. If it does not contain $S_{B(p)}$, this means that $S_{B(p)}$ is a leader, or that the subgroup of $\text{OPT}_{B(p)}$ is of the form $(\{\ell_0\}, \{\text{OPT}_{B(p)}\})$, for some center $\ell_0 \in S_0$. If $S_{B(p)}$ is a leader of a group of size larger than $2/\varepsilon$, then $S_{B(p)}$ is not in any subgroup and is therefore not in any set $X$ of any subgroup $(X, Y)$ and the above property will thus be satisfied since the swap pairs are obtained from the subgroups. Otherwise, if $S_{B(p)}$ is in a group of size smaller than $2/\varepsilon$, then merge the subgroup of $S_{B(p)}$ with the subgroup of $\text{OPT}_{B(p)}$. If $S_{B(p)}$ is not a leader, then $\text{OPT}_{B(p)}$ is in a subgroup $g' = (\{\ell_0\}, \{\text{OPT}_{B(p)}\})$. In which case, merge the subgroup $g'$ with the subgroup containing $S_{B(p)}$.

A *swap* pair $X, Y$ is thus created for the subgroups resulting from the above merge process. By the definition of the process, if $S_{B(p)}$ is in $X$, then $\text{OPT}_{B(p)} \in Y$. Moreover, since for each subgroup $(A, B)$ we have $|A| = |B|$ the resulting process guarantees that the resulting swap pairs $(X, Y)$ are such that $|X| = |Y|$. This ensures that the solution $S_0 - X \cup Y$ is feasible for any swap pair $X, Y$ hence created. Next, to use local optimality, or in other words argue that the cost of any solution $S_0 - X \cup Y$ is at least as large as $(1 - \varepsilon/k)$ times the cost of $S_0$, we have to show that the merge process did not lead to subgroups $X, Y$ such that $|X| + |Y| \leq 12/\varepsilon$.

Observe first that subgroups of size 2, e.g., subgroups of the form $(\{\ell'\}, \{\text{OPT}_{B(p)}\})$, are merged with at most one other group, i.e., the group containing $S_{B(p)}$. Hence this part of the operation only yields swaps of size at most $6/\epsilon$. Next, we argue that each subgroup of size larger than 2 is only merged with at most one other subgroup. Indeed, if a subgroup $(X_i, Y_i)$ of size larger than 2 is merged with another subgroup $(X_j, Y_j)$, it means that the leader is a facility $S_{B(p)}$ for some anchor point $p$ and $Y_j$ contains $\text{OPT}_{B(p)}$. However, note that the leader $\ell'$ of $(X_j, Y_j)$ must be in the ball of radius $\frac{\gamma}{2}\delta(p)$ by definition of the swaps and the bipartite graph and so closer to $p$ than to any other anchor point by the correctness of Algorithm 1. Therefore, the only subgroup that can be merged to the group containing $S_{B(p)}$ is the subgroup containing $\text{OPT}_{B(p)}$ and the sizes of the groups created is at most twice larger than the subgroup sizes.

From the above discussion, we have that each swap pair $(X, Y)$ is feasible, so by local optimality, we have $\text{cost}(S_0 - X \cup Y) \geq (1 - \varepsilon/k)\text{cost}(S_0)$, since otherwise the algorithm would have made another iteration and replaced $S_0$ with $S_0 - X \cup Y$. To conclude the analysis of the cost of $S_0$, we bound the cost of the solution $S_0 - X \cup Y$ for each swap pair $(X, Y)$, adapting the analysis of Gupta and Tangwongsan [2008]—here one can immediately adapt their analysis to obtain the desired bounds for $k$-means. To do so, $\forall a \in A$, let $\text{OPT}^a$ (respectively $S_0^a$) denote the center of OPT (respectively $S_0$) that is the closest to $a$. Observe that we have the following properties for the collection of swap pairs defined above:

1. For each swap pair $(X, Y)$, for each optimal center $\text{OPT}^a$, let $r^a$ be the center of $S_0$ closest to $\text{OPT}^a$. If $\text{OPT}^a$ is not in $Y$, then there is a center $\ell \notin X$ at distance at most $\max(\text{dist}(\text{OPT}^a, r^a), \frac{4}{\gamma - 2}\text{dist}(\text{OPT}^a, r^a))$ from $\text{OPT}^a$.

   Indeed, this comes from the fact that each center $\text{OPT}^a$ adjacent to a center $\ell$ of $S_0$ in the bipartite graph satisfies, for any swap pair $(X, Y)$, if $\ell \in X$ then $\text{OPT}^a \in Y$ (and by the contrapositive, if $\text{OPT}^a \notin Y$ then $\ell \notin X$). Now, observe that either $\ell$ is $r^a$ (namely the center

of $S_0$ closest to $\mathrm{OPT}^a$), or in the case where $\mathrm{OPT}^a$ is an anchored center of an anchor point $p$, $\ell$ is the closest center of $S_0$ in the ball of radius $\frac{\gamma}{2}\delta(p)$ centered at $p$. In this case, the closest center $\mathrm{OPT}^a$ in $S_0$ is by triangle inequality at distance at least $(\frac{\gamma}{2}-1)\delta(p)$ from $p$. Moreover, since $S_0$ is feasible, there is a center of $S_0$ at distance at most $\delta(p)$ from $p$, and by triangle inequality at most $2\delta(p)$ from $\mathrm{OPT}^a$. Thus, $\mathrm{dist}(\mathrm{OPT}^a, \ell) \leq 2\delta(p) \leq \frac{4}{\gamma-2}\mathrm{dist}(\mathrm{OPT}^a, r^a)$ as claimed (recall $\gamma > 2$).

2. Each center of OPT appears in exactly one pair $(X, Y)$.

3. Each center of $S_0$ appears in expectation in $(1 + \varepsilon)$ pairs $(X, Y)$. This is because when the leader of a group is substituted randomly with another element of the group, each point of the group has probability $\varepsilon$ of being used as substitute and to appear in two pairs $(X, Y)$ (and probability $(1 - \varepsilon)$ of appearing in exactly one pair).

For each swap pair $(X, Y)$, for each point $a \in A$, we bound $\Delta(a, (X, Y)) = \mathrm{dist}(a, S_0 - X \cup Y) - \mathrm{dist}(a, S_0)$.

For the swap $(X, Y)$ where $\mathrm{OPT}^a \in Y$, we have $\mathrm{OPT}^a \in S_0 - X \cup Y$ and so the contribution of $a$ to the cost of $S_0 - X \cup Y$ is at most $\mathrm{OPT}(a)$. We thus have $\Delta(a, (X, Y)) \leq \mathrm{OPT}(a) - S_0(a)$.

For the swaps $(X, Y)$ where $S_0^a \in X$ and $\mathrm{OPT}^a \notin Y$. For a fixed point $a$, the number of such swaps is in expectation at most $(1 + \varepsilon)$ by the above discussion. For such a swap, since $\mathrm{OPT}^a \notin Y$, the cost of $a$ is at most the distance from $a$ to the center $\ell$ of $S_0$ that is at distance at most $\max(\mathrm{dist}(\mathrm{OPT}^a, r^a), \frac{4}{\gamma-2}\mathrm{dist}(\mathrm{OPT}^a, r^a))$ from $\mathrm{OPT}^a$ and that is not in $X$ by Property 1 above.

By triangle inequality, the distance from $a$ to $\ell$ is at most

$$\mathrm{dist}(a, \mathrm{OPT}^a) + \mathrm{dist}(\mathrm{OPT}^a, \ell)$$

$$\leq \mathrm{dist}(a, \mathrm{OPT}^a) + \max\left(1, \frac{4}{\gamma-2}\right)\mathrm{dist}(\mathrm{OPT}^a, r^a)$$

$$\leq \mathrm{dist}(a, \mathrm{OPT}^a) + \max\left(1, \frac{4}{\gamma-2}\right)[\mathrm{dist}(a, S_0^a) + \mathrm{dist}(a, \mathrm{OPT}^a)],$$

since $r^a$ is the center of $S_0$ that is the closest to $\mathrm{OPT}^a$. We thus have $\Delta(a, (X, Y)) \leq \mathrm{OPT}(a) + \max(1, \frac{4}{\gamma-2})(S_0(a) + \mathrm{OPT}(a)) - S_0(a)$.

For the remaining swaps, the contribution of $a$ is $S_0(a)$. $\Delta(a, (X, Y)) \leq 0$.

Next define $m_\gamma = \max(1, \frac{4}{\gamma-2})$. Thus, for each point $a \in A$, we have

$$\mathbb{E}[\sum_{(X,Y)} \Delta(a, (X, Y))] \leq (1 + \varepsilon)(2 + m_\gamma)\mathrm{OPT}(a) - S_0(a)(2 + \varepsilon - (1 + \varepsilon)m_\gamma),$$

where the expectation is over the random choices made to define the swaps. Thus,

$$\sum_{a \in A} \mathbb{E}[\sum_{(X,Y)} \Delta(a, (X, Y))]$$

$$\leq \sum_{a \in A}(1 + \varepsilon)(2 + m_\gamma)\mathrm{OPT}(a) - (2 + \varepsilon - (1 + \varepsilon)m_\gamma)S_0(a).$$

By local optimality, we have for each $(X, Y)$, $\sum_{a \in A} \Delta(a, (X, Y)) > -\varepsilon\mathrm{cost}(S_0)/k$. Summing the above inequality over all swaps, we conclude $-\varepsilon\mathrm{cost}(S_0) \leq \sum_{a \in A}(1 + \varepsilon)(2 + m_\gamma)\mathrm{OPT}(a) - (2 + \varepsilon - (1 + \varepsilon)m_\gamma)S_0(a)$ since the number of swaps is at most $k$.

Therefore the ratio is at most $(1 + O(\varepsilon))\frac{2 + \max(1, \frac{4}{\gamma-2})}{2 - \max(1, \frac{4}{\gamma-2})}$, for $\gamma > 4$, and small enough $\varepsilon$, as desired. $\square$

## 4   Fast algorithm

In this section we focus on the $k$-means problem and we show how to modify the local-search algorithm presented in Lattanzi and Sohler [2019] to obtain a bicriteria approximation for our

problem, Theorem 1.2. The key intuition is to use the concept of anchor zones introduced in the previous section to allow only the swaps that preserve our fairness guarantees.

---

**Algorithm 3:** Scalable algorithm for individually fair $k$-means

---

**Require:** $X, k, Z, \gamma$
1: $C \leftarrow \emptyset, S_0 \leftarrow \emptyset$
2: $S_0 \leftarrow \text{SEEDING}(X, \delta(\cdot), \gamma)$.
3: Define each point $p \in S_0$ as an *anchor point*, and the ball $B(p)$ of radius $\gamma\delta(p)$ around $p$ as an *anchor zone*.
4: Let $T \subseteq X \setminus S_0$ be a set of $k - |S_0|$ randomly selected points
5: $S \leftarrow S_0 \cup T$
6: **for** $i \leftarrow 2, 3, \ldots, Z$ **do**
7:     $S \leftarrow \text{CONSTRAINEDLOCALSEARCH++}(X, S, B(\cdot))$
8: **return** $S$

---

---

**Algorithm 4:** CONSTRAINEDLOCALSEARCH++

---

**Require:** $X, S, B(\cdot)$
1: Sample $p \in X$ with probability $\frac{\text{cost}(\{p\}, S)}{\sum_{q \in X} \text{cost}(\{q\}, S)}$
2: $Q \leftarrow \{q \in S | \forall x \in S_0 : (S \setminus \{q\} \cup \{p\}) \cap B(x) \neq \emptyset\}$
3: $q^* \leftarrow \arg\min_{q \in Q} \text{cost}(X, S \setminus \{q\} \cup \{p\})$
4: **if** $\text{cost}(X, S \setminus \{q^*\} \cup \{p\}) < \text{cost}(X, S)$ **then**
5:     $S \leftarrow S \setminus \{q^*\} \cup \{p\}$
6: **return** $S$

---

For simplicity of exposition in this section we consider the classic setting where $A = C = X$.

Toward this end, we need to change both the initialization and the swapping procedure of the local-search algorithm to take into account the *radius constraints*. As for initialization, as in the previous section, we first add a new center as long as there exists a point $p$ at distance greater than $\gamma\delta(p)$ from the current set of centers. We refer to the obtained set of centers as $S_0$. If $|S_0|$ is larger than $k$, then we know that the input is infeasible; otherwise we add additional points as centers until we obtain a set of $k$ centers $S$. We say that a point is an *anchor point* if it is in $S_0$. Furthermore we define the ball $B(p)$ of radius $\gamma\delta(p)$ centered at $p$ as the *anchor zone* for $p$.

As for the swaps, we select a random point $q$ using $D^2$-sampling as in Lattanzi and Sohler [2019]. If there is a subset $S'$ obtained by swapping an element of $S$ with $q$, such that (i) $|S'| = k$, (ii) every anchor zone contains at least one point in $S'$, and (iii) $\sum_{p \in X} \text{dist}(p, C)^2 > \sum_{p \in X} \text{dist}(p, S')^2$, then we change our current solution from $S$ to $S'$. Interestingly we show that after $O(k \log n\Delta)$ iterations, the solution will have constant-factor expected approximation for cost and moreover it violates the radius constraints by at most a factor of $2\gamma$. See the pseudocode in Algorithm 3.

Now we show that our algorithm obtains a constant bicriteria approximation for individually fair $k$-means[3]. Our proof uses many ingredients of the proof in Lattanzi and Sohler [2019] with careful modifications to handle the additional constraints imposed by the algorithm[4]. In the remaining part of this section we prove our main theorem focusing on the novel part of our proofs.

### 4.1 Analysis (Proof of Theorem 1.2)

As in Lattanzi and Sohler [2019], the main observation behind our proof is that every step of our algorithm in expectation reduces the solution cost by a factor $O\left(1 - \frac{1}{k}\right)$. Considering that the cost

---

[3]In this section we do not optimize to obtain a small approximation factor; the main advantage of this algorithm is its very practical running time. Nevertheless we show in Section 5 that our algorithm is more efficient and obtains higher-quality result compared to prior work.

[4]An interesting open question is to use the more recent analysis of LOCALSEARCH++ by Choo et al. [2020] to improve the running time of our algorithm to $O(dnk^2)$. However, it is not clear how to obtain the necessary strong guarantees similar to the one in Lemma 12 of Choo et al. [2020].

of the initial solution is at most $\Delta^2 n$, this implies that $O(k \log n\Delta)$ iterations suffice to obtain a constant approximation.

To simplify the exposition we assume that every cluster in the optimal solution has non-zero cost.[5]

Next we state two lemmas outlining the algorithm's analysis. Their proofs use ideas similar to those in Lattanzi and Sohler [2019], and are deferred to supplementary materials.

**Lemma 4.1.** *Let $X$ be the set of points from a feasible instance, $\gamma \geq 3$, and $S$ a set of centers with cost $cost(X, S) > 2000 Opt_k$. With probability $\frac{1}{1000}$, for $S' = \text{CONSTRAINEDLOCALSEARCH}++(X, S)$, we have $cost(X, S') \leq (1 - \frac{1}{100k})cost(X, S)$.*

**Lemma 4.2.** *Let $X$ be the set of points from a feasible instance, and $\hat{S}$ a set of centers with $cost(X, \hat{S}) \leq \gamma n\Delta(X)^2$. After running $Z \geq 200000 k \log(\gamma n \Delta(X))$ rounds of Algorithm 4 on $\hat{S}$ outputs a solution $S$ such that $E[cost(X, S)] \in O(OPT_k)$.*

Here we show how to use the two lemma to prove Theorem 1.2.

*Proof of Theorem 1.2.* The algorithm returns infeasible only if it finds $k + 1$ disjoint individual fairness balls. But in that case, the problem is infeasible (their fairness constraints cannot be satisfied with $k$ points).

Let $\hat{S}$ be the set $S$ before calling CONSTRAINEDLOCALSEARCH++. In this set, every point $p$ has distance at most $\gamma\delta(p)$ from a center so $cost(X, \hat{S}) \leq \gamma n(\Delta(X))^2$. Lemma 4.2 then shows that after $Z$ calls to CONSTRAINEDLOCALSEARCH++, we obtain a constant approximation.

Now we show that at any point in time during the execution of the algorithm $\max_{p \in X} dist(p, S) \leq 2\gamma\delta(p)$. The algorithm guarantees to keep at least one point in every anchor ball. Moreover every point $p$ is at distance at most $\gamma\delta(p)$ from an anchor point $p'$ with $\delta(p) > \delta(p')$. The anchor ball $B(p')$ must have a center $c \in S$, so $dist(c, p') \leq \gamma\delta(p')$. Thus by triangle inequality $dist(c, p) \leq 2\gamma\delta(p)$.

It takes $O(dnk)$ time to compute the initial set $\hat{S}$. In order to implement the local search, we need to compute the cost of swapping the new sample point with an old center. This requires to iterate over all clusters and for each cluster we need to compute the distance to all other centers and to check that there is at least one center in each anchor ball. Thus, a local search step requires $O(dkn + dk)$ time in the worst case, which leads to an overall running time of $O(dnkZ)$. The Theorem follows. $\quad\square$

## 5 Empirical analysis

In this section we evaluate empirically the algorithms presented and we compare them with state-of-the-art methods from the literature. First, we describe our empirical methodology, and then we provide the experimental results.

In our analysis, all datasets used are *publicly available*. We implemented our algorithms, as well as the other baselines in Python, and we ran each instance of our experiments independently on a standard single-core machine. We will make the code available as open-source before publication.

**Datasets.** We used several real-world datasets from the UCI Repository Dheeru and Karra Taniskidou [2017] that are standard in the clustering literature. This includes: adult Kohavi et al. [1996] $n = 32561, d = 6$, bank Moro et al. [2014] $n = 45211, d = 3$, diabetes Dheeru and Karra Taniskidou [2017] $n = 101766, d = 2$ skin Bhatt and Dhall [2010], $n = 245057, d = 4$, shuttle[6] $n = 58000, d = 9$, and covertype Blackard and Dean [1999], $n = 581012, d = 54$. For consistency with prior work, for adult, bank and diabetes we use the same set of columns used in the analysis of Mahabadi and Vakilian [2020]. We preprocess each dataset to have zero mean and unit standard deviation in every dimension. All experiments use the Euclidean distance.

**Algorithms.** We consider the following algorithms.
– **ICML20** Mahabadi and Vakilian [2020]: We implemented the algorithm following the recommendation of the paper (i.e., using a single swap in the local search and a factor $3\delta(p)$ instead of $6\delta(p)$ in

---

[5]Note that this is w.l.o.g., since we can artificially increase the cost of every cluster by adding for each point a copy at infinitesimal distance.

[6]Thanks to NASA for releasing the dataset.

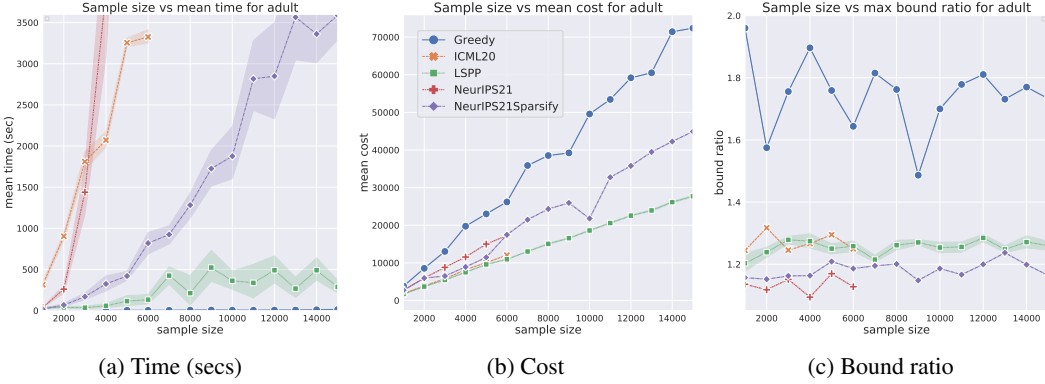

Figure 1: Mean completion time, cost, and bound ratio for the algorithms on adult dataset subsampled to different sizes, $k = 10$. The shades represent the $95\%$ confidence interval (notice that some algorithms are deterministic). Runs that did not complete in 1 hour are not reported.

the initialization). We set $\epsilon = 0.01$ in the algorithm.

– **NeurIPS21** Chakrabarty and Negahbani [2021]: We use the Python code provided by the authors.[7] We use both the more accurate algorithm **NeurIPS21** and the faster algorithm using sparsification (**NeurIPS21Sparsify**).

– **Greedy**: Similarly to prior work we consider the execution of the greedy seeding algorithm as a baseline.

– **LSPP**: We implemented our local-search algorithm with modifications similar to that of ICML20 (a single swap and $\mu = 3$ factor in seeding algorithm). We also modified the algorithm to run only a fixed number of local-search iterations (namely 500) in all experiments. Moreover, we added at the end of the algorithm the execution of an algorithm similar to Lloyd's method, and as is typical, we repeat for 20 iterations the following procedure: We assign each point to the nearest center. Then we obtain the mean of the clusters. Notice that the mean minimizes the $k$-means cost, but it may not be a feasible solution for the distance bound. For this reason, the next center is obtained by approximating the closest feasible point to the mean, on the line between the current center and the mean. Though this procedure does not alter the theoretical guarantees, it improves the results empirically.

**Metrics.** We focus on three key metrics: the $k$-means cost of clustering, the average runtime of algorithm and the *bound ratio* $\max_p \frac{\text{dist}(p,S)}{\delta(p)}$ where $S$ is the solution of the algorithm. We repeat each experiment configuration 10 times and report the mean and standard deviation of the metrics.

**Comparison with other baselines.** In this section we report a comparison of our algorithm with the other baselines. For all experiments, unless otherwise specified, we replicate the setting of individual fairness Mahabadi and Vakilian [2020] for $\delta(p)$, by setting $\delta(p)$ as the distance to the $n/k$-th nearest point.

Notice that the ICML20 algorithm evaluates, for each iteration of local search, all possible swaps of one center with a non-center while NeurIPS21 and NeurIPS21Sparsify both depend on computing all-pairs distances in $O(dn^2)$ time. This makes these algorithms not scalable to large datasets, unlike our algorithm. Therefore all prior experiments Mahabadi and Vakilian [2020], Chakrabarty and Negahbani [2021] used a subsample of $\approx$1000 elements from the datasets to run their algorithm. In this section we use a similar approach for the sake of consistency.

In Figure 1a, 1b, 1c, we report the results of the various algorithms for different sizes of the sample on the adult dataset, fixing $k = 10$. We allowed each algorithm to run for up to 1 hour. Notice in Figure 1a, how our algorithm is orders of magnitude faster than the baselines even for very small sample sizes. Even the faster NeurIPS21Sparsify variant is still much slower than our algorithm.

Moreover, while the running time of all baselines (except simple Greedy) increases significantly with size, our algorithm scales much better and has a running time similar to the naive Greedy baseline.

---

[7]The code was obtained from `https://github.com/moonin12/individually-fair-k-clustering` and adapted.

Then we focus on the $k$-means cost of the solution in Figure 1b. Notice that our algorithm (LSPP) has a cost better than that of all baselines. Finally, Figure 1c shows the max ratio of distance of a point $p$ to centers vs $\delta(p)$. While ICML20 and LSPP have statistically comparable bounds (significantly better than their worst case guarantees), NeurIPS21 and NeurIPS21Sparsify have slightly better bounds.

**Effect of $k$.** A similar overall picture appears in Figure 2a, 2b, where we report the results of the various algorithms for different $k$'s on a sample of 1000 elements in the adult dataset. Notice the statistically significant improvement in cost in Figure 2a and the comparable or slightly higher bound ratio in Figure 2b. The results observed before are confirmed in all datasets, as shown in Table 1, where we report the cost and bound ratio for all datasets, subsampled to 1000 elements, and $k = 10$.

**Experiments on the full datasets.** The scalability of our algorithm allows us to run it on the full datasets with up to $1/2$ million elements[8], orders of magnitude larger than the datasets used in prior work. In this section, we run our algorithm and the fast Greedy baseline on all complete datasets, using $k = 10$. Our algorithm completed all runs in less than 60 minutes.

In order to compare with all the slower baselines we allow ICML20, NeurIPS21 and NeurIPS21Sparsify to run on a subsample of the data containing 4000 points (but we evaluate the solution on the entire dataset). This of course has no theoretical guarantee and can perform especially poorly in case of outliers.

For this large-scale experiment, the input bound $\delta(p)$ for each point $p$ is set using the $n/k$-th closest point in a random sample of 1000 elements. In all but one dataset, our algorithm has a significantly lower $k$-means cost than that all other baselines. Similarly to above results, our algorithm has similar or better ratio bound than that of ICML20 (with the sampling heuristic), while the ratio bound of NeurIPS21 and NeurIPS21Sparsify is sometimes lower. In any instance our algorithm has much better ratio that the worst-case theoretical guarantees. The results are reported in the supplementary materials.

| dataset | algorithm | $k$-means cost | bound ratio |
|---|---|---|---|
| adult | Greedy | 3832.6 (—) | 2.0 (—) |
| | ICML20 | 1854.9 (—) | 1.2 (—) |
| | NeurIPS21 | 2744.3(—) | **1.1** (—) |
| | NeurIPS21Sparsify | 2745.5 (—) | 1.2 (—) |
| | LSPP | **1726.0** (9.4) | 1.2 (0) |
| bank | Greedy | 1081.8 (—) | 1.8 (—) |
| | ICML20 | 568.4 (—) | 1.4 (—) |
| | NeurIPS21 | 784.2 (—) | **1.2** (—) |
| | NeurIPS21Sparsify | 761.2 (—) | **1.2** (—) |
| | LSPP | **515.6** (7.3) | 1.4 (0.1) |
| covtype | Greedy | 50629.8 (—) | 1.3 (—) |
| | ICML20 | 42121.5 (—) | 1.1 (—) |
| | NeurIPS21 | 47810.6 (—) | 1.1(—) |
| | NeurIPS21Sparsify | 46078.6 (—) | 1.1 (—) |
| | LSPP | **36080.8** (632) | **1.0** (0.0) |
| diabetes | Greedy | 522.3 (—) | 2.1 (—) |
| | ICML20 | 267.0 (—) | **1.1** (—) |
| | NeurIPS21 | N/A | N/A |
| | NeurIPS21Sparsify | 299.8 (—) | **1.1** (—) |
| | LSPP | **243.9** (8.5) | 1.2 (0.2) |
| shuttle | Greedy | 2647.4 (—) | 2.0 (—) |
| | ICML20 | 1335.0 (—) | 1.8 (—) |
| | NeurIPS21 | 2494.8 (—) | **1.0**(—) |
| | NeurIPS21Sparsify | 2477.1 (—) | 1.2 (—) |
| | LSPP | **1219.1** (31.3) | 1.8 (0.1) |
| skin | Greedy | 584.3 (—) | 2.7 (—) |
| | ICML20 | **292.8**(—) | 2.3 (—) |
| | NeurIPS21 | 379.4 (—) | **1.1** (—) |
| | NeurIPS21Sparsify | 384.1 (—) | **1.1** (—) |
| | LSPP | 325.1 (19.4) | 2.6 (0.3) |

Table 1: Cost and max bound ratio for all datasets subsampled for 1000 elements and $k = 10$ (stddev in parentheses for the LSPP randomized algorithm). N/A indicates that the algorithm did not complete in 1 hour.

The less scalable algorithms still provide decent results via sampling the dataset. This will not be necessarily the case though. In particular, sampling will inevitably remove *underrepresented* groups. Consider a dataset with two (or more) slices corresponding to different countries or groups, with similar characteristics yet different populations. If the subsets appear in different parts of the space, the sampling procedure will fail to take into account (many features from) the smaller countries or groups, defeating the purpose of these algorithms—individual fairness.

## 6 Conclusions

After improving state of the art for individually fair clustering (in terms of theoretical cost and fairness guarantees), we present a scalable local-search algorithm that, despite inferior theoretical guarantees, performs very well in practice.

---

[8]A implementation in C++ would scale this up further, but we wanted to be consistent in comparison to prior work.

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

# A   Proof of Section 4

The proof in this section follow closely the structure of the proofs in Lattanzi and Sohler [2019] with some modification to carefully handle the anchor zones constraints.

## A.1   Proof of Lemma 4.2

*Proof.* By Lemma 4.1 we know that if before any call of CONSTRAINEDLOCALSEARCH++ the cost of the centers is bigger than $2000\text{Opt}_k$ then with probability $\frac{1}{1000}$ we reduce the cost by a $\left(1 - \frac{1}{100k}\right)$ multiplicative factor.

Now consider another random process $Y$ with initial value equal to $\text{cost}(X, \hat{S})$, which for $Z = 100000k \log n\Delta(X)^2$ iterations, it reduces the value by a $\left(1 - \frac{1}{100k}\right)$ multiplicative factor with probability $1/1000$, and finally increases the value by an additive $2000\text{Opt}_k$. It is not hard to see that the final value of $Y$ stochastically dominates the cost of the solution our algorithm produces. So the final expected value of $Y$ is larger than the expected value of $\text{cost}(X, S)$ conditioned on the initial clustering $\hat{S}$. Furthermore,

$$E[Y] = 2000\text{Opt}_k + \text{cost}(X, \hat{S})\cdot$$
$$\sum_{i=0}^{Z} \binom{Z}{i} \left(\frac{1}{1000}\right)^i \left(\frac{999}{1000}\right)^{Z-i} \left(1 - \frac{1}{100k}\right)^i$$
$$= \text{cost}(X, \hat{S}) \left(1 - \frac{1}{100000k}\right)^Z$$
$$+ 2000\text{Opt}_k$$
$$\leq \frac{\text{cost}(X, \hat{S})}{n\Delta(X)^2} + 2000\text{Opt}_k.$$

This implies that $E[\text{cost}(X, S)|\hat{S}] \leq \frac{\text{cost}(X,\hat{S})}{n\Delta(X)^2} + 2000\text{Opt}_k$. Our upper-bound on the cost of $\hat{S}$ is deterministic, hence $E[\text{cost}(X, S)] \leq \frac{\text{cost}(X,\hat{S})}{n\Delta(X)^2} + 2000\text{Opt}_k \leq 2001\text{Opt}_k$. □

## A.2   Proof of Lemma 4.1

Before proving the lemma we recall two well-known results. The following lemma is folklore:

**Lemma A.1.** *Let $X \subseteq \mathbb{R}^d$ be a set of points and let $c \in \mathbb{R}^d$ be a center. Then we have $\text{cost}(X, \{c\}) = |X| \cdot \|c - \mu(X)\|^2 + \text{cost}(X, \mu(X))$.*

We will also use the following lemma (rephrased from Corollary 21 in Feldman et al. [2018]).

**Lemma A.2.** *Let $\epsilon > 0$. Let $p, q \in \mathbb{R}^d$ and let $C \subseteq \mathbb{R}^d$ be a set of $k$ centers. Then $|\text{cost}(\{p\}, C) - \text{cost}(\{q\}, C)| \leq \epsilon \cdot \text{cost}(\{p\}, C) + (1 + \frac{1}{\epsilon})\|p - q\|^2$.*

We assume that the optimal solution $S^* = \{c_1^*, \ldots, c_k^*\}$ is unique (this can be enforced using proper tie breaking) and use $X_1^*, \ldots, X_k^*$ to denote the corresponding optimal partition. We will also use $S = \{c_1, \ldots, c_k\}$ to refer to our current clustering with corresponding partition $X_1, \ldots, X_k$. When the indices are not relevant, we will drop the index and write, for example, $c \in S$.

We use notations and proof strategy similar to Kanungo et al. [2002]. We start by partitioning the optimal centers into *anchor centers*, $A^*$, and *unconstrained centers*, $U^*$. An optimal center is in $A^*$ if it is the closest optimal center to an anchor point (breaking ties arbitrarily), the remaining centers form the set of unconstrained centers. We say that an optimal center $c^* \in U^*$ is *captured* by a center $c \in S$ if $c$ is the nearest center to $c^*$ among all centers in $S$. Also we say that an optimal center $c^* \in A^*$ with corresponding anchor point $a$ is captured by a center $c \in S$ if $c$ is the nearest center to $c^*$ among all centers in the anchor zone defined by $a$. Note that a center $c \in S$ may capture more than one optimal center and every optimal center is captured by exactly one center from $S$ (ties are broken arbitrarily). Some center in $c$ may not capture any optimal center. Similarly to Kanungo et al. [2002] we call these centers *lonely* and we denote them with $L$. Finally, let $H$ be the index set

of centers capturing exactly one cluster. W.l.o.g., we assume that for $h \in H$ we have that $c_h \in S$ captures $c_h^* \in S^*$, i.e., the indices of the clusters with a one-to-one correspondence are identical.

Note that the above definition is slightly different from the classic definition in Kanungo et al. [2002]. In fact, an optimal center may not be captured by its closest center but by its closest center in the anchor zone. Nevertheless we can show that it is still possible to recover a similar result to the one in Lattanzi and Sohler [2019] in this setting.

Note one useful proposition of our definition.

**Proposition A.3.** *Let $c^* \in A^*$ be an optimal center with corresponding anchor point $a$, and let $c'$ be the closest point in $S$ to $c^*$, and let $c^*$ be captured by the center $c \in S$. Then $dist(c^*, c) \leq$ $\mu+1/\mu-1 dist(c^*, c')$*

*Proof.* If $c'$ is within distance $\mu\delta(a)$ to $a$, the lemma follows from $dist(c^*, c) = dist(c^*, c')$ by definition of $c$ and anchor ball. Otherwise we know that $c^*$ is at distance at most $\delta$ from $a$, $c$ is at distance at most $\mu\delta$ from $a$, and $c'$ is at distance at least $\mu\delta$ from $a$. The lemma follows from the triangle inequality. □

We will use the above definition as in Lattanzi and Sohler [2019]. Intuitively, if a center $c$ captures exactly one cluster of the optimal solution, we think of it as a candidate center for this cluster. In this case, if $c$ is far away from the center of this optimal cluster, we argue that with good probability we sample a point close to the center. In order to analyze the change of cost, we will argue that we can assign all points in the cluster of $c$ that *are not in the captured optimal cluster* to a different center without increasing their contribution by too much. This will be called the reassignment cost and is formally defined in the definition below. We will show that with good probability we sample from a cluster such that the improvement for the points in the optimal cluster is significantly bigger than the reassignment cost.

If a center is lonely, we think of it as a center that can be moved to a different cluster. Again, we will argue that with high probability we can sample points from other clusters such that the reassignment cost is much smaller than the improvement for this cluster.

Now we start to analyze the cost of reassignment of the points due to a center swap.

We would like to argue that reassigning the points currently assigned to a cluster center with index from $H$ or $L$ to other clusters is small. As discussed above, for $h \in H$, we will assign all points from $X_h$ that are not in $X_h^*$ to other centers. For $l \in L$ we will consider the cost of assigning all points in $X_i$ to other clusters. We use the following definition to capture the cost of this reassignment introduced in Lattanzi and Sohler [2019].

**Definition A.4.** Let $X \subseteq \mathbb{R}^d$ be a point set and $S \subseteq \mathbb{R}^d$ be a set of $k$ cluster centers and let $H$ be the subset of indices of cluster centers from $S = \{c_1, \ldots, c_k\}$ that capture exactly one cluster center of an optimal solution $S^* = \{c_1^*, \ldots, c_k^*\}$. Let $X_i, X_i^*, 1 \leq i \leq k$, be the corresponding clusters. Let $h \in H$ be an index with cluster $X_h$ and w.l.o.g. let $X_h^*$ be the cluster in the optimal solution captured by $c_h$. The reassignment cost of $c_h$ is defined as

$$\text{reassign}(X, S, c_h) = \text{cost}(X \setminus X_h^*, S \setminus \{c_h\}) - \text{cost}(X \setminus X_h^*, S).$$

For $\ell \in L$ we define the reassignment cost of $c_\ell$ as

$$\text{reassign}(X, S, c_\ell) = \text{cost}(X, S \setminus \{c_\ell\}) - \text{cost}(X, S).$$

We will now prove the following bound on reassignment costs. We note that this proof is similar to the proof in Lattanzi and Sohler [2019] but it includes key differences to handle the fact that optimal centers may not be assigned to the closest center in the current solution.

**Lemma A.5.** *For $r \in H \cup L$ we have*

$$reassign(X, S, c_r) \leq \frac{13}{100} cost(X_r, S) + 332 cost(X_r, S^*).$$

*Proof.* We only present the case $r \in H$. The case $r \in L$ is almost identical (in fact, simpler). We observe that $\text{reassign}(X, S, c_r) = \text{cost}(X_r \setminus X_r^*, S \setminus \{r\}) - \text{cost}(X_r \setminus X_r^*, S)$ since vertices in clusters other than $X_r$ will still be assigned to their current center. If $r \in H$, we assign every point

in $X_r \cap X_i^*$, $i \neq r$, to the center that captured the center of $X_i^*$. While this assignment may not be optimal, its cost provides an upper bound on the cost of reassigning the points: We move every point in $X_r \cap X_i^*$, $i \neq r$, to the center of $X_i^*$. Now the closest center of $S$ to these points is a center with distance close to the one that captured the center of $X_i^*$, which, for points not in $X_r^*$, cannot be $r$, since $r$ is in $H$. The fact that the squared moved distance of each point equals its contribution to the optimal solution allows us to get an upper bound on the cost change using Lemma A.2. After this, we move the points back to their original location while keeping their cluster assignments fixed. Again we can use the bound on the overall moved distance together with Lemma A.2 to obtain a bound on the change of cost. Combining the two gives an upper bound on the increase of cost that comes from reassigning the points. Details follow.

Let $Q_r$ be the (multi)set of points obtained from $X_r \setminus X_r^*$ by moving each point in $X_i^* \cap X_r$, $i \neq r$, to $c_i^*$. We apply Lemma A.2 with $\epsilon = 1/100$ to get an upper bound for the change of cost with respect to $S$ that results from moving the points to $Q_r$. For $p \in X_r \setminus X_r^*$ let $q_p \in Q_r$ be the point of $Q_r$ to which $p$ has been moved. We have:

$$|\text{cost}(\{p\}, S) - \text{cost}(\{q_p\}, S)|$$
$$\leq \quad \frac{1}{100}\text{cost}(\{p\}, S) + 101 \cdot \text{cost}(\{p\}, S^*).$$

Summing up over all points in $X_r \setminus X_r^*$ yields

$$|\text{cost}(X_r \setminus X_r^*, S) - \text{cost}(Q_r, S)|$$
$$\leq \quad \frac{1}{100}\text{cost}(X_r \setminus X_r^*, S) + 101 \cdot \text{cost}(X_r \setminus X_r^*, S^*).$$

Let $Q_{r,i}$ be the points in $Q_r$ that are nearest to center $c_i \in S$ and let $X_{r,i}$ be the set of their original locations. For $p \in X_{r,i}$ that has been moved to $q_p \in Q_{r,i}$ with $q_p$ let $c_i'$ be the closest point to $q_p$ not equal to $r$. Note that the only case in which $c_i \neq c_i'$ is when $c_i = r$. Furthermore, $q_p$ is not captured by $r$ because $r$ captures $c_r^*$ and is in $H$. So by Proposition A.3 we know $\text{cost}(\{q_p\}, \{c_i'\}) \leq \frac{\mu+1}{\mu-1}\text{cost}(\{q_p\}, \{c_i\})$. Thus we have:

$$|\text{cost}(\{q_p\}, \{c_i\}) - \text{cost}(\{p\}, \{c_i'\})|$$
$$= \quad |\text{cost}(\{q_p\}, \{c_i\}) - \text{cost}(\{q_p\}, \{c_i'\})|$$
$$+ \quad |\text{cost}(\{q_p\}, \{c_i'\}) - \text{cost}(\{p\}, \{c_i'\})|$$
$$\leq \quad \frac{2}{\mu-1}\text{cost}(\{q_p\}, \{c_i\}) + \frac{1}{100}\text{cost}(\{q_p\}, \{c_i'\})$$
$$+ \quad 101 \cdot \text{cost}(\{p\}, \{q_p\})$$
$$\leq \quad \frac{1}{\mu-1}\left(2 + \frac{\mu+1}{100}\right)\text{cost}(\{q_p\}, \{c_i\})$$
$$+ \quad 101 \cdot \text{cost}(\{p\}, \{q_p\}),$$

where in the first inequality we used Lemma A.2 with $\epsilon = 1/100$.

Summing up over all points in $X_r \setminus X_r^*$ and the corresponding points in $Q_r$ yields

$$\left|\text{cost}(Q_r, S) - \sum_i \text{cost}(X_{r,i}, S \setminus \{r\})\right|$$
$$\leq \quad \frac{1}{\mu-1}\left(2 + \frac{\mu+1}{100}\right)\text{cost}(Q_r, S) +$$
$$101 \cdot \text{cost}(X_r \setminus X_r^*, S^*)$$
$$\leq \quad \frac{26}{25}\left(\frac{11}{100}\text{cost}(X_r \setminus X_r^*, S) +\right.$$
$$\left. 101 \cdot \text{cost}(X_r \setminus X_r^*, S^*)\right) +$$
$$101 \cdot \text{cost}(X_r \setminus X_r^*, S^*)$$
$$\leq \quad \frac{3}{25}\text{cost}(X_r, S) + 231 \cdot \text{cost}(X_r, S^*),$$

where the second inequality uses the bound on $\mu \geq 3$. Hence,

$$
\begin{aligned}
\text{reassign}&(X, S, c_r) \\
&= |\text{cost}(X_r \setminus X_r^*, S) - \sum_i \text{cost}(X_{r,i}, S \setminus \{r\}| \\
&\leq |\text{cost}(X_r \setminus X_r^*, S) - \text{cost}(Q_r, S)| + \\
&\quad\ |\text{cost}(Q_r, S) - \sum_i \text{cost}(X_{r,i}, S \setminus \{r\})| \\
&\leq \frac{13}{100}\text{cost}(X_r, S) + 332\text{cost}(X_r, S^*). \qquad \square
\end{aligned}
$$

Now that we have a good bound on the reassignment cost we make a case distinction. Recall that we assume that for every $h \in H$ the optimal center captured by $c_h$ is $c_H^*$, i.e., the indices are identical. We first consider the case that $\sum_{h \in H} \text{cost}(X_h^*, S) > \frac{1}{3}\text{cost}(X, S)$.

With the previous lemma at hand, we can focus on the centers $h \in H$ where replacing $h$ by an arbitrary point close to the optimal cluster center of the optimal cluster captured by $h$ improves the cost of the solution significantly. As in Lattanzi and Sohler [2019] we call such clusters *good* and make this notion precise in the following definition.

**Definition A.6.** A cluster index $h \in H$ is called *good* if

$$
\text{cost}(X_h^*, S) - \text{reassign}(X, S, c_h) - 9\text{cost}(X_h^*, \{c_h^*\}) > \\
\frac{1}{100k} \cdot \text{cost}(X, S).
$$

The above definition estimates the gain of replacing $c_h$ by a point close to the center of $X_h^*$ by considering a clustering that reassigns the points in $X_h$ that do not belong to $X_h^*$ and assigns all points in $X_h^*$ to the new center. Now we want to show that we have a good probability to sample a good cluster. In particular, we first argue that the sum of cost of good clusters is large. We note that the following proof is a simple adaptation of Lattanzi and Sohler [2019].

**Lemma A.7.** *If* $3 \sum_{h \in H} cost(X_h^*, S) > cost(X, S) \geq 2000 Opt_k$, *then*

$$
\sum_{h \in H, h \text{ is good}} cost(X_h^*, S) \geq \frac{9}{400} cost(X, S).
$$

*Proof.* We have $\sum_{h \in H} \text{cost}(X_h^*, C) \geq \frac{1}{3}\text{cost}(X, S)$ and by the definition of good and Lemma A.5

$$
\begin{aligned}
\sum_{h \in H, h \text{ is not good}} \text{cost}(X_h^*, S) \quad &\leq \quad \sum_{h \in H} \text{reassign}(X, S, c_h) + \\
&\qquad 9\text{Opt}_k + \frac{1}{100}\text{cost}(X, S) \\
&\leq \quad \frac{14}{100}\text{cost}(X, S) + 341\text{Opt}_k.
\end{aligned}
$$

Using that $\text{cost}(X, S) \geq 2000\text{Opt}_k$ we obtain that

$$
\sum_{h \in H, h \text{ is not good}} \text{cost}(X_h^*, S) \leq \frac{621}{2000} \cdot \text{cost}(X, S).
$$

So $\sum_{h \in H, h \text{ is good}} \text{cost}(X_h^*, S) \geq \frac{9}{400} \cdot \text{cost}(X, S)$. The lemma follows. $\qquad \square$

Now we present a lemma from Lattanzi and Sohler [2019] that whenever a cluster has high cost w.r.t. $C$, it suffices to consider the points close to the optimal center to get an approximation of the cost of the cluster. We will then use this fact to argue that we sample with good probability a point close to the center.

**Lemma A.8** (Lemma 6 from Lattanzi and Sohler [2019] restated). *Let $Q \subseteq \mathbb{R}^d$ be a point set and let $S \subseteq \mathbb{R}^d$ be a set of $k$ centers and let $\alpha \geq 9$. If $cost(Q, S) \geq \alpha \cdot cost(Q, \{\mu(Q)\})$ then*

$$cost(R, S) \geq \left(\frac{\alpha - 1}{8}\right) \cdot cost(Q, \{\mu(Q)\}),$$

*where $R \subseteq Q$ is the subset of $Q$ at squared distance at most $\frac{2}{|Q|} \cdot cost(Q, \{\mu(Q)\})$ from $\mu(Q)$.*

Now we can argue that sampling according to sum of squared distances will provide us with constant probability with a good center. Consider any index $h \in H$ with $h$ being good. We will apply Lemma A.8 with $Q = X_h^*$ and $\alpha = cost(Q, S)/cost(Q, \mu(Q))$. Note that by the definition of good, we have that $\alpha \geq 9$. Now let us define $R_h^*$ to be the set $R$ guaranteed by Lemma A.8. We have $cost(R_h^*, S) \geq \frac{\alpha - 1}{8} cost(X_h^*, \{c_h^*\}) = \frac{\alpha - 1}{8\alpha} cost(X_h^*, S) \geq \frac{1}{9} cost(X_h^*, S)$ by our choice of $\alpha$ (observe that $c_h^*$ equals $\mu(X_h^*)$). Since the sum of squared distances of points in good clusters is at least $9/400 cost(X, S)$ by Lemma A.7, we conclude that $\sum_{h \in H, h \text{ is good}} cost(R_h^*, S) \geq \frac{9}{9 \cdot 400} cost(X, S)$. Thus, the probability to sample a point from $\sum_{h \in H, h \text{ is good}} cost(R_h^*, S)$ is more than $1/400$. By the definition of good, if we sample such a point $c \in R_h^*$ we can swap it with $c_h$ to get a new clustering of cost at most $cost(X, S \setminus \{c_h\} \cup \{c\}) \leq cost(X, S) - cost(X_h^*, S) + \text{reassign}(X, S, \{c_h\}) + cost(X_h^*, \{c\})$. By Lemma A.1 we know that $cost(X_h^*, \{c\}) \leq 9cost(X_h^*, \{c_h^*\})$. Hence, with probability at least $1/400$ the new clustering has cost at most

$$cost(X, S) - (cost(X_h^*, S) - \text{reassign}(X, H, c_h)$$
$$-9cost(X_h^*, \{c_h^*\})$$
$$\leq (1 - \frac{1}{100k}) \cdot cost(X, S).$$

To check that the swap is feasible we only need to make sure that the swap is feasible if $\{c_h^*\} \in A^*$. Otherwise we already that the anchor balls are covered by other centers. If $\{c_h^*\} \in A^*$, let $a$ be the anchor point corresponding to $c_h^*$. Note that from the definition of good cluster that $cost(X_h^*, S) - 9cost(X_h^*, \{c_h^*\}) > 0$ so by Lemma A.1 we have $d(\{c_h^*\}, S) \geq 9d(\{c_h^*\}, c)$. So given that the radius of the anchor ball is $3\delta(a)$ and the distance between $c_h^*$ and $a$ is bounded by $\delta(a)$ by triangle inequality we have that $c$ is inside the anchor ball. This proves our lemma in the first case.

In the second case, we have $\sum_{h \in H} cost(X_h^*, S) < 1/3 cost(X, S)$. Now let $R = \{1, \ldots, k\} \setminus H$, so we get $\sum_{r \in R} cost(X_r^*, S) \geq 2/3 cost(X, S)$. Observe that $R$ equals the index set of optimal cluster centers that were captured by centers that capture more than one optimal center. This is because every optimal center is captured by one center and $R$ does not include $H$. In this case, if the index of a center of our current solution is in $R \setminus L$ we cannot easily move the cluster center without having impact on other clusters. What we do instead is to use the centers in $L$ as candidate centers for a swap. Note that those swaps are always feasible because inside each anchor ball we have also a center not in $L$. Similar to the case above we will argue that we can swap a center from $L$ with a point that is close to an optimal center of a cluster $X_r^*$ for some $r \in R$.

Recall that we have already bounded the cost of reassigning a center in $L$ so we just need to argue that the probability of sampling a good center is high enough.

In particular, we focus on the centers $r \in R$ and swap an arbitrary center $\ell \in L$ with an arbitrary point close to one of the centers in $R$ to improve the cost of the solution. Slightly overloading notation, we call such cluster centers *good* and make this notion precise in the following definition.

**Definition A.9.** A cluster index $i \in \{1, \ldots, k\}$ is called *good*, if there exists a center $\ell \in L$ such that

$$cost(X_i^*, S) - \text{reassign}(X, S, \ell) - 9cost(X_i^*, \{c_i^*\}) >$$
$$\frac{1}{100k} \cdot cost(X, S).$$

The above definition estimates the cost of removing $\ell$ and inserting a new cluster center close to the center of $X_i^*$ by considering a clustering that reassigns the points in $X_i^*$ and assigns all points in $X_i^*$ to the new center. In the following we will now argue that the sum of cost of good clusters is large, this will be useful to show that the probability of sampling such a cluster is high enough.

**Lemma A.10.** *If* $3\sum_{h\in H} cost(X_h^*, S) \leq cost(X, S)$ *and* $cost(X, S) \geq 2000Opt_k$ *we have*

$$\sum_{r\in R, r \text{ is good}} cost(X_r^*, S) \geq \frac{1}{20} cost(X, S).$$

*Proof.* We have $\sum_{r\in R} cost(X_r^*, S) \geq 2/3 cost(X, S)$. Note that $|R| \leq 2|L|$. By the definition of good and Lemma A.5

$$\sum_{r\in R, r \text{ is not good}} cost(X_r^*, S)$$

$$\leq 2|L| \min_{\ell\in L} \text{reassign}(X, S, \ell) + 9Opt_k$$

$$+\frac{1}{100} cost(X, S)$$

$$\leq 2\sum_{\ell\in L} \text{reassign}(X, S, \ell) + 9Opt_k$$

$$+\frac{1}{100} cost(X, S)$$

$$\leq \frac{27}{100} cost(X, S) + 673Opt_k.$$

Using that $\sum_{i\in\{1,...,k\}} cost(X_i^*, S) \geq 2000Opt_k$ we obtain that

$$\sum_{r\in R, r \text{ is not good}} cost(X_r^*, S) \quad \leq \quad \frac{1213}{2000} cost(X, S)$$

Now the bound follows by combining the previous inequality with $\sum_{r\in R} cost(X_r^*, S) \geq 2/3 cost(X, S)$. $\square$

Note that also in this case we can now argue similarly as in the other case that sampling according to sum of squared distances will provide us with constant probability with a good center using Lemma A.8. In fact, since the sum of squared distances of points in good centers is at least $1/20 cost(X, S)$ by Lemma A.10, it follows together with Lemma A.8 that we sample a point from a good cluster $X_r^*$ that is within distance two times the average cost of the cluster with probability $\frac{1}{1000}$. By the definition of good, we know that such a point improves the cost of the current clustering by at least a factor of $(1 - \frac{1}{100k})$. Thus, Lemma 4.1 follows.

# B   Additional experimental results

As we mentioned before, our algorithm is the only one that runs on the big datasets within reasonable time (and memory).

In order to compare with all the slower baselines we allow ICML20, NeurIPS21 and NeurIPS21Sparsify to run on a subsample of the data containing 4000 points (but we evaluate the solution on the entire dataset). This of course has no theoretical guarantee and can perform especially poorly in case of outliers.

For this large-scale experiment, the input bound $\delta(p)$ for each point $p$ is set using the $n/k$-th closest point in a random sample of 1000 elements.

The results in Table 2 shows that in all but one dataset, our algorithm has a significantly lower $k$-means cost than that all other baselines. Similarly to above results, our algorithm has similar or better ratio bound than that of ICML20 (with the sampling heuristic), while the ratio bound of NeurIPS21 and NeurIPS21Sparsify is sometimes lower. In any instance our algorithm has much better ratio that the worst-case theoretical guarantees.

**Standard deviation of the metrics in large datasets.**   In Table 3 we report the standard deviation for the metrics in Table 2. Notice that in this experiment, the input to ICML20, NeurIPS21 and

| dataset | algorithm | $k$-means cost | bound ratio |
|---|---|---|---|
| adult | Greedy | 1.56E+05 | 1.8 |
| | ICML20 | 6.59E+04 | 1.4 |
| | NeurIPS21 | 1.14E+05 | **1.2** |
| | NeurIPS21Sparsify | 1.02E+05 | **1.2** |
| | LSPP | **6.14E+04** | 1.4 |
| bank | Greedy | 8.57E+04 | 1.9 |
| | ICML20 | 3.23E+04 | 1.6 |
| | NeurIPS21 | 5.68E+04 | **1.2** |
| | NeurIPS21Sparsify | 5.70E+04 | **1.2** |
| | LSPP | **3.02E+04** | 1.6 |
| covtype | Greedy | 3.33E+07 | 1.3 |
| | ICML20 | 2.84E+07 | **1.1** |
| | NeurIPS21 | 2.76E+07 | **1.1** |
| | NeurIPS21Sparsify | 2.80E+07 | **1.1** |
| | LSPP | **2.50E+07** | **1.1** |
| diabetes | Greedy | 6.60E+04 | 2.7 |
| | ICML20 | 3.00E+04 | 1.3 |
| | NeurIPS21 | N/A | N/A |
| | NeurIPS21Sparsify | 3.36E+04 | **1.2** |
| | LSPP | **2.66E+04** | 1.4 |
| shuttle | Greedy | 4.89E+05 | 2.3 |
| | ICML20 | 1.91E+05 | 2.0 |
| | NeurIPS21 | 2.60E+05 | **1.0** |
| | NeurIPS21Sparsify | 2.72E+05 | 1.1 |
| | LSPP | **1.79E+05** | 2.1 |
| skin | Greedy | 1.80E+05 | 2.1 |
| | ICML20 | **7.47E+04** | 1.8 |
| | NeurIPS21 | 9.36E+04 | **1.1** |
| | NeurIPS21Sparsify | 1.03E+05 | **1.1** |
| | LSPP | 9.27E+04 | 3.1 |

Table 2: Mean Cost and max bound ratio for all full-sized datasets and k=10 with ICML20, NeurIPS21 and NeurIPS21Sparsify ran on a sample of 4000 points.

| dataset | algorithm | cost stddev | bound ratio stddev |
|---|---|---|---|
| adult | ICML20 | 6.81E+02 | 1.00E-01 |
| | NeurIPS21 | 1.34E+04 | 0.00E+00 |
| | NeurIPS21Sparsify | 7.65E+03 | 0.00E+00 |
| | LSPP | 8.84E+02 | 0.00E+00 |
| bank | ICML20 | 1.04E+03 | 1.00E-01 |
| | NeurIPS21 | 5.95E+03 | 0.00E+00 |
| | NeurIPS21Sparsify | 6.57E+03 | 0.00E+00 |
| | LSPP | 6.85E+02 | 1.00E-01 |
| covtype | ICML20 | 2.23E+05 | 0.00E+00 |
| | NeurIPS21 | 1.85E+05 | 0.00E+00 |
| | NeurIPS21Sparsify | 3.83E+05 | 0.00E+00 |
| | LSPP | 4.51E+05 | 0.00E+00 |
| diabetes | ICML20 | 8.19E+02 | 2.00E-01 |
| | NeurIPS21 | N/A | N/A |
| | NeurIPS21Sparsify | 1.22E+03 | 1.00E-01 |
| | LSPP | 9.58E+02 | 1.00E-01 |
| shuttle | ICML20 | 1.25E+04 | 2.00E-01 |
| | NeurIPS21 | 6.57E+03 | 0.00E+00 |
| | NeurIPS21Sparsify | 1.37E+04 | 0.00E+00 |
| | LSPP | 1.05E+04 | 3.00E-01 |
| skin | ICML20 | 3.51E+03 | 3.00E-01 |
| | NeurIPS21 | 1.60E+03 | 1.00E-01 |
| | NeurIPS21Sparsify | 1.31E+04 | 0.00E+00 |
| | LSPP | 4.80E+02 | 2.00E-01 |

Table 3: Standard deviation of cost and max bound ratio for all full-sized datasets and k=10 with ICML20, NeurIPS21 and NeurIPS21Sparsify ran on a sample of 4000 points .

NeurIPS21Sparsify algorithms are run on a random subsample, so this makes the algorithm non-deterministic. Notice that our algorithm has statistically significantly lower cost than the other baselines in almost all datasets.

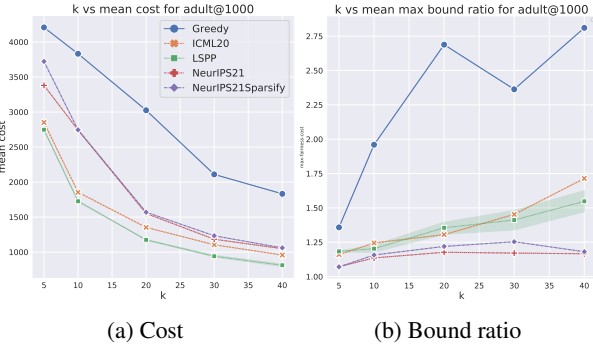

(a) Cost                              (b) Bound ratio

Figure 2: Mean completion cost and bound ratio for the algorithms on adult dataset subsampled to 1000 elements and different $k$'s. The shades represent the $95\%$ confidence interval.

