# OpenReview forum: "Scalable and Improved Algorithms for Individually Fair Clustering"
_NeurIPS.cc/2022/Workshop/TSRML — TSRML2022_

### Official Review · Reviewer_61SB · 2022-10-18

**Overall Recommendation:** This work has a solid contribution in…
**Overall Rating:** 7

**Summary:**

In this paper, the authors focus on the individually fair clustering problem where there is a relatively equal chance for every instance to be aligned with a cluster. The existing algorithms are mostly computationally inefficient due to a search procedure, thus the authors proposed to use anchor zones for a more efficient constrained local search. The authors provided detailed theoretical analysis and appealing experimental results to compare the proposed method with baseline approaches.

**Strengths:**

-Overall, the paper is well organized, and the authors are introducing their work, including the problem set-up and theoretical analysis, in detail.
-The contribution is well-motivated, as the existing fair clustering algorithms seem to have a big computational issue.
-The theoretical analysis is neat. Although I didn't check through the proof, it seems the theorems can well support the advantages of the proposed algorithm.


**Weaknesses:**

I am wondering whether and how the proposed method can deal withing a varying number of clusters (k) and the weights of the clusters, just as [Vakilian and Yalçiner 2022] did.


**Review Confidence:**

3: The reviewer is fairly confident that the evaluation is correct

---

### Official Review · Reviewer_nHGe · 2022-10-20
**Strong Experiments**

**Overall Rating:** 6

**Summary:**

The article proposes an approximated algorithm for individually fair k-median (k-means and k-center) that matches state-of-the-art approaches in terms of guarantees on the cost and fairness of the obtained clusters. On the other hand, it can drastically reduce the running time of the previous algorithms by applying a randomized contained local search approach. The results on several standard UCI datasets demonstrate the time efficiency of the algorithm enabling it to be used for large-scale problems despite the competitors.

**Strengths:**

- The literature review is one of the main strengths of the paper, and the authors have connected the current work to the related papers very well.
- The proofs are correct to the best of the reviewer's knowledge.
- The experiments work surprisingly well in terms of final cost function despite the fast randomized method proposed in Algorithm 4.  Moreover, the algorithm reduces the total runtime error drastically.

**Weaknesses:**

- The proofs are based on the previous works, which is not a problem by itself. However, it is unclear in what sense the new proposed algorithm is superior to the previous ones. The bounds for cost and fairness of the obtained solution existed in the previous works. On the other hand, Lemma 3.2 is significant in terms of runtime efficiency. However, it requires a large number of restarts to guarantee to obtain an optimal solution. If the algorithm is implemented based on Lemma 3.2 (with so many restarts), is it still efficient? And if not, how can you theoretically ensure Algorithm 4 finds an optimal solution?

- It might not be realistic to assume that a completely fair solution satisfies the $\delta$-distance constraint for all points in the dataset, especially when the dataset is large-scale (high number of samples). Can the work be extended to the setting where a small proportion of data points can actually violate the fairness constraint?

**Overall Recommendation:**

I think it is necessary to address the aforementioned weaknesses before accepting the paper. While the numerical experiments look promising, the theoretical results have limited novelty.

**Review Confidence:**

4: The reviewer is confident but not absolutely certain that the evaluation is correct

---

### Decision · Program_Chairs · 2022-10-23

Accept